Subject Area:
molecular biology/structural biology/ biochemistry/biophysics/genetics

Keywords:
model fungi, mutational analysis, fluorescent microscopy, phylogeny, molecular dynamics, transporter structure

Author for correspondence:
George Diallinas
e-mail: diallina@biol.uoa.gr

# Tales of tails in transporters

Emmanuel Mikros[1] and George Diallinas[2]

[1]Faculty of Pharmacy, National and Kapodistrian University of Athens, Panepistimioupolis, 15771 Athens, Greece
[2]Department of Biology, National and Kapodistrian University of Athens, Panepistimioupolis, 15781 Athens, Greece

GD, 0000-0002-3426-726X

Cell nutrition, detoxification, signalling, homeostasis and response to drugs, processes related to cell growth, differentiation and survival are all mediated by plasma membrane (PM) proteins called transporters. Despite their distinct fine structures, mechanism of function, energetic requirements, kinetics and substrate specificities, all transporters are characterized by a main hydrophobic body embedded in the PM as a series of tightly packed, often intertwined, α-helices that traverse the lipid bilayer in a zigzag mode, connected with intracellular or extracellular loops and hydrophilic N- and C-termini. Whereas longstanding genetic, biochemical and biophysical evidence suggests that specific transmembrane segments, and also their connecting loops, are responsible for substrate recognition and transport dynamics, emerging evidence also reveals the functional importance of transporter N- and C-termini, in respect to transport catalysis, substrate specificity, subcellular expression, stability and signalling. This review highlights selected prototypic examples of transporters in which their termini play important roles in their functioning.

## 1. Transporters and their tails

Transporters are membrane proteins that mediate the selective passage of nutrients, metabolites or drugs across cellular membranes. Their activity is essential for cell survival, division and differentiation and consequently for the life of all organisms. This is reflected in the high number of corresponding genes in all genomes (approx. 5–15%) and in several associated genetic or other diseases caused by transporter malfunction (e.g. cystic fibrosis, diabetes, neurodegeneration, etc.). The great majority of solute transporters are either facilitators, not requiring energy for downhill transport, or secondary active transporters, which need energy coupling provided by the co-transport (symport) or exchange (antiport) of ions or other solutes down their electrochemical gradients. Two other types of transmembrane proteins involved in transport, ATP-dependent primary active transporters and ion channels, are significantly distinct in structure, function and evolution from secondary active transporters and facilitators. This review will discuss aspects concerning the roles of cytosolic tails specifically in secondary active transporters.

Secondary active transporters consist of 10–14 hydrophobic or amphipathic mostly α-helical transmembrane segments (TMSs), connected through hydrophilic loops (L), and mostly hydrophilic N- or C-terminal regions of variable length, which in the majority of cases are cytoplasm facing. These transporters bind their substrates at a single binding site from one side of the membrane and transport it to the opposite side by a translocation mechanism that requires significant reversible conformational alterations. Surprisingly, transporters of distant evolutionary families and of distinct function and energetic requirement might share a common architectural fold, one among the four currently identified, namely the LeuT, the major facilitator superfamily (MFS), the GltPh and the NhaA fold [1–3]. Notably, however, while different folds are associated with distinct mechanisms of transport, known as the rocking-bundle (LeuT), rocker-switch (MFS) or elevator-like sliding (GltPh, NhaA and distantly related

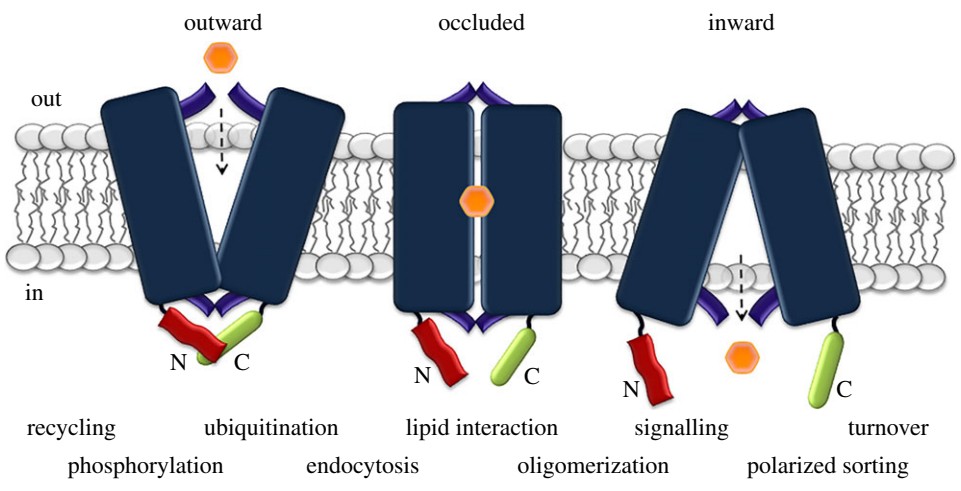

royalsocietypublishing.org/journal/rsob    Open Biol. **9**: 190083

**Figure 1.** Cytosolic N- and C-termini play a crucial role in transporter expression, function and turnover. The figure highlights in a simplified manner how the topology of cytosolic termini alters depending on the overall conformation of the transporters (i.e. outward facing, occluded, inward facing) and this altered conformation is crucial for regulating both intramolecular events (i.e. transport activity, allosteric regulation) and intermolecular interactions (i.e. ubiquitination, endocytosis, sorting, signalling, etc.). Notice that in the outward-facing conformation the N- and C-tails are in closer contact with other domains of the transporter than in the inward-facing conformation. For more details see main text.

LeuT-like transporters), the function of all transporters is based on a common mechanism which drives the alteration from an *outward-* to an *inward*-facing topology [4–9]. This common mechanism is elicited by substrate binding, translocation and release. In addition to this major conformational alteration imposed by substrate binding and release, *gating* by specific short segments or residues at the entrance or exit, or along the substrate translocation trajectory, controls substrate access to and from the major binding site and thus finely regulates transport [1,10,11]. Overall, the current view is that transporters alternate in at least five structurally distinct conformations during transport: outward-facing open (substrate reception), outward-facing occluded (substrate oriented in the major binding site and closure of the outer gate), fully occluded (substrate stabilized in a major binding site), inward-facing occluded (substrate binding induces inward conformation) and inward-facing open (inner gate opens and substrate released) [6–9]. A relatively novel concept concerning transporters is that substrate *specificity* is not solely determined by the interactions taking place in the major binding site. Instead, genetic, biochemical and structural evidence showed that specificity is mostly regulated via proper functioning of the gates or other amino acid residues that are located along the substrate translocation trajectory, and which do not constitute part of the bona fide major substrate binding site [10–17].

The generally accepted idea that substrate recognition and transport are carried out by synergy of specific transmembrane segments, and also by specific dynamic loops, left an open question of what the role is of domains that seem not to be involved, at least directly, in transport catalysis. A prominent case is that of the N- and C-termini of transporters. These are, in the majority of cases, cytoplasmic. Unlike functional domains essential for transport activity, transporter termini are usually extremely variable with respect to their length and amino acid composition, even among paralogues or orthologues that have practically identical transport activity and specificity (G. Diallinas 2019, unpublished observations originating from multiple sequence alignments; see also http://www.tcdb.org/). In several cases termini have been shown to be important for recruiting cytoplasmic effectors involved in subcellular sorting, targeting, stability, cell-specific modification or regulated turnover, or to be critical for homo-oligomerization and interaction with membrane lipids, and thus indirectly essential for transporter function. Termini have also been shown to be critical for transport activity and substrate specificity via allosteric regulation of the transport mechanism. These emerging functional 'tales' of transporter termini are the subject of this review (figure 1).

## 2. Evidence that transporter tails have important functional roles

When primary amino acid sequences of homologous transporters are compared using standard multiple sequence alignments it becomes apparent that eukaryotic transporters have, in general, longer termini than prokaryotic ones. In fact, in some families, prokaryotic members have practically no or only very short termini of a few amino acids. This is exemplified by members of the two largest, ubiquitously conserved and well-characterized transporter families, the amino acid–polyamine–organocation (APC) superfamily and the MFS [18]. Below we describe selected cases of members of the APC and APC-related families.

Eukaryotic APC proteins, in general, possess significantly longer N- and C-terminal hydrophilic regions than their bacterial homologues. APC termini are little conserved, except for some short motifs shared among close homologues. However, experimental work has revealed that specific elements, which might not always be well conserved, are indeed critical for APC function. For example, the N-termini of yeast APCs possess relatively short motifs and specific Lys residues necessary for regulated turnover via ubiquitination, endocytic internalization and sorting in endosomes and vacuoles [19–21]. The C-terminus of yeast APCs, which in general seems more conserved than the N-terminus, is predicted to form specific secondary structures and also contains motifs critical for sorting, turnover, palmitoylation or partitioning to plasma membrane (PM) microdomains [21–26]. Most surprisingly, the termini of specific APC-related transporters control transport activity and substrate specificity via

allosteric interactions with other cytosolic loops [27,28]. Selected examples of N- or C-tails of APC transporters that play functional roles will be discussed in more detail later.

Two nucleobase-specific transporter families structurally related to APC [29] provided evidence that cytosolic tails have important functional roles. These are the NCS1 (nucleobase cation symporter 1; [30,31]) and the NAT (nucleobase ascorbate transporters, also known as NCS2; [32–36]) families. In NCS1, the N-termini of both prokaryotic and eukaryotic members contain a well-conserved motif conforming to the consensus **N**-X-**D/S**-$\Phi$-X-**P** (absolutely conserved residues in bold, $\Phi$ is a hydrophobic amino acid). Despite the fact that the prokaryotic homologues have relatively shorter termini than the eukaryotic ones, all NCS1 members possess this motif at the same distance (approx. 10 amino acid residues) from TMS1, as delimited by crystallography or molecular modelling of members of the family. Since this sequence has remained conserved for millions of years of evolution it might be an element serving a basic biochemical property of transporters, such as proper folding or transport activity or both. This sequence proved to be redundant for transporter folding or function [31,37], but surprisingly was found to be critical for finely determining substrate specificities in NCS1 transporters ([31]; see below). Unlike the N-terminus, the C-tail of members of NCS1 does not include any ubiquitously conserved motif. However, the acidic sequence **E/D**-X-**E**-**E** has been shown to be absolutely essential for endocytic regulation via specific ubiquitination of the purine-related FurE transporter of the fungus *Aspergillus nidulans* [17]. Versions of such short poly-acidic motifs are present in the tails of different transporters, but their positioning related to the last TMS or the C-end differs. It will be interesting to investigate whether this element acts as an autonomous functional motif determining the endocytic sensitivity of different NCS1 paralogues.

In contrast to NCS1, the termini of NAT proteins do not contain ubiquitously conserved motifs, but rather are variable in length and amino acid composition [11,13]. An apparent redundancy of NAT termini for transport activity is supported by that fact that several prokaryotic members possess practically no or only extremely short terminal segments (e.g. 2–4 residues; [34]). In simple eukaryotes, such as slime moulds and protists, the N- and C-termini of NATs (15–50 and 28–91 residues, respectively) are definitely longer than in prokaryotes, but extremely variable, containing no prominent conserved motifs. In fungi, the NAT N- and C-termini are 71–120 and 15–70 residues long, respectively, and, interestingly, the N-terminus contains a nearly absolutely conserved sequence. This motif (**G**-**D**-**Y/F**-**D**-**Y/W/F**), which is not present in NATs of prokaryotes, slime moulds, protists, plants or metazoa, has been shown to be essential for proper endoplasmic reticulum (ER) exit and PM sorting, and might also be critical for oligomerization of UapA, a uric acid–xanthine NAT transporter in *A. nidulans* [11,38]. The C-terminus of fungal NATs has no absolutely conserved motifs, but in dikarya (higher fungi) a well-conserved di-acidic motif **E**-X-**E** (or in its more extended version E/D-E/D-X-E/D-E/D) is present just neighbouring the TMS14 (last membrane segment). In the case of the UapA this motif proved absolutely essential for ubiquitination and endocytosis in response to various physiological or stress signals [33,39]. This motif has been proposed to be essential for recognition of UapA by the $\alpha$-arrestin

adaptor–ubiquitin ligase complex ArtA/HulA, which is responsible for ubiquitination of a specific Lys (K572) residue in the distal part of the C-tail [39]. In line with these findings, deletion of the entire C-terminal region of UapA leads to total block in endocytosis in response to various physiological or stress signals [40], and notably does not affect transport activity and kinetics. In plants, the N-terminus of NATs can be quite variable in different phylogenetic sub-clades, ranging in length from 33 to 240 amino acids. In the most abundant and canonical sub-clade of plant NATs (i.e. the one that includes characterized nucleobase transporters and known functional motifs within specific TMSs) the N-tail (33–50 residues) includes the well-conserved motif **D/E**-**Q**-**L/**$\Phi$-X-X-$\Phi$-X-**Y**-**C**-**I**-X-S, just upstream of TMS1, while the C-terminus has two conserved motifs, **D/E**-**R**-**G**-X-X-**W**-**W**, 5–8 residues downstream from TMS14, and **D/E**-X-**R/A**-X-X-**E**-**F**-**Y**-X-**L**-**P**-X$_6$-**F**, at the most distal segment of the C-tail. No studies have addressed the role and functional importance of these motifs.

In metazoa the cytosolic termini of NATs are of variable lengths, ranging from 12 to 142 residues for the N-terminus, and from 28 to 98 for the C-terminus. In general, shorter tails are found in the most primitive animals, while fish, amphibians and mammals have the longest. No N-terminal conserved motif is shared by all animals or even among evolutionary related groups (i.e. among fish, amphibia, reptiles, birds, arthropods, mammals, etc.) or even among related sub-groups (e.g. insects, primates, etc.). Only true orthologues share significant similarity in their cytosolic tails, but even this is not always the case. For example, the L-ascorbate mammalian transporters of the SVCT2 group have quite conserved N-termini, while their functionally very similar paralogue group, SVCT1, shows significant sequence variation in its tails. This difference might be related to their distinct tissue- or cell-specific expression profiles, which also seem to be controlled by distinct subcellular sorting pathways. In line with this, the N-terminus of SVCT2 seems also critical for redirecting apical SVCT1 to the basolateral membrane [41,42]. The C-terminus of metazoan NATs is more conserved than the N-tail, including three sequence motifs. The first is the absolutely conserved sequence **E/Q**-**R**-**G**-$\Phi$-X-X-**W**, located 5 residues downstream from TMS14. In all vertebrates, except fish, this motif is even more conserved, being extended to E-E-R-G-$\Phi$-X-X-W. The second motif conforms to the sequence **Y**-**D/N**-X-**P**-**F**-**G**-$\Phi$, found 27–37 residues downstream from TMS14. The third motif is the sequence $\Phi$-$\Phi$-**P**-$\Phi$/**F**-X-**P**, 48–59 residues distal from TMS14. Noticeably, the last motif is less conserved in a divergent clade of NAT, the SVCT3 group, which includes members of unknown specificity, rather than being nucleobase or L-ascorbate transporters. We predict that these motifs will be important for the stability, turnover and/or the subcellular trafficking of metazoan NAT transporters.

Another APC-related family which presents an interest in respect to the role of its cytosolic termini is the neurotransmitter sodium symporter (NSS) family. NSS transporters include several biomedically important transporters, such as the well-studied dopamine (DAT) and serotonin (SERT) transporters, all conforming to the LeuT-like 5+5 inverted repeat fold [6,43]. This family has members in all major animal groups, but distant members also exist in prokaryotes and fungi. The N-terminus of NSS, which is variable in length (21–179 residues) and amino acid sequence, does not show any

ubiquitous motif. However, there are very well-conserved motifs in transporters with similar substrate specificity in evolutionarily closely related groups. For example, there is a well-conserved sequence, **P-K-E-V-E-L-I-L-V-K-E-Q/H-N-G-V-Q-F/Y-T,** in DAT-like homologues of mammals, fish, amphibians, birds and reptiles (i.e. in vertebrates), 16 residues upstream from TMS1. This motif is however absent in other chordates (e.g. *Ciona intestinalis*) and protostomes (e.g. arthropods, insects, etc.). The C-terminus of NSS (40–45 residues) is in general more conserved than the N-terminus, and contains the motif A-I-**Y-K**-$X_4$-**P-G**-X-**Φ**-X-**D/E/Q**-K/R-X$_7$-**P** just downstream (2 residues) from TMS12. This motif does not seem to be directly related to specificity as it is present in several animal transporters recognizing distinct neurotransmitters (e.g. DAT, SERT or glycine, less so in betaine transporters). In the DAT of all vertebrates, the most C-terminal segment is also extremely well conserved, including the sequence **L-Φ-X-X-G-X-V-R-Q-F-X-L-X-W-W-L** (where most X are polar residues). Additionally to the presence of short sequence motifs, experimental evidence or *ab initio* structural predictions have suggested that the N- and C-termini of NSS contain partially conserved folds that seem to be extremely important for function, via their interaction with each other and cytosolic loops of the main body of the NSS transporters [44–50]. The case of the DAT transport tails is discussed in more detail later. The above-described terminal motifs and other sequences experimentally proven to have an important function in transporters are summarized in table 1.

## 3. Tails are essential for ER exit and sorting of transporters to the PM

Eukaryotic transporters, being polytopic membrane proteins, are co-translationally translocated from ribosomes into the ER membrane, via the translocon complex [51,52]. The direct translocation in the hydrophobic environment of membrane lipids of the ER enables de novo transporters to fold properly and acquire their functional conformation. The process of co-translational translocation coupled with concurrent folding provides an early quality control point, as misfolded membrane proteins are retained within the ER membrane and elicit mechanisms that lead to their degradation, such as ER-associated degradation (ERAD) [53–55] or chaperone-mediated selective autophagy [56]. In all cases, the ribosomes which synthesize transporters (or other membrane proteins) need to attach to the translocon complex in the ER (or the PM in prokaryotes). Evidence has suggested that the first translated TMS of polytopic membrane proteins interacts with the signal recognition particle (SRP) and guides the ribosome to the translocon [57,58]. Subsequently, other TMSs seem to also act as signals for proper folding during ER translocation [57,59]. Truncated versions of transporters, missing TMS1 or several other TMSs, might also be inserted into the ER, but never exit from it, and are often degraded by ERAD [60,61]. It thus seems that while entering the ER might not require specific cytosolic signals, but only the presence of hydrophobic segment(s), exiting from the ER is much more demanding, as it requires specific multivalent interactions of transporter cargoes with proteins of the ER exit machinery (e.g. the COPII coat complex) and specific membrane lipids, necessary to promote ER membrane curvature, formation and release of coated secretory vesicles [62–65]. It is more

than obvious that such interactions necessitate proper transporter folding, which in several cases is also necessary for proper homo-oligomerization [38,66] and partitioning into specific lipid bilayer domains [67], processes important for ER exit. Additionally, ER exit, at least of some transporters, also requires the presence of specific, autonomous or context-dependent, sequence motifs, present nearly exclusively in cytosolic termini. As will be shown below, transporter termini might have important roles in ER exit directly or indirectly.

ER exit sequence motifs are essential for the recognition of transmembrane proteins by specific components of an ER exit mechanism, basically by the COPII adaptor coat protein Sec24 or its paralogues [68–71]. Sec24-recognized ER export signals are usually variable short sequences, such as di-acidic (**D/E**-X-**D/E**), hydrophobic and aromatic (**FF, YY, LL, FY, ΦXΦXΦ**), or other more variable short motifs [72–78]. Notably, some similar short motifs are also recognized by AP-1 and AP-2 adaptor complexes that regulate clathrin assembly at the trans-Golgi network (TGN) or the PM, respectively, or by the AP-3 adaptor complex that is implicated in endosome to lysosome trafficking [79]. A **D-I-D** tripeptide located in the C-terminus of the general amino acid transporter Gap1 of *Saccharomyces cerevisiae*, just next to the last TMS (TMS12), was probably the first motif in transporters shown to be necessary for loading into COPII vesicles [80]. It should be noted that, in several cases where mutations that map in TMSs or internal loops lead to ER retention, this seems to be an indirect effect due to partial misfolding, rather than because they define bona fide ER exit signals [56,81].

The involvement of specific chaperones for ER exit has also been supported in several transporter families [82–84]. However, no conserved sequence motif in the cytosolic termini of transporters has been shown to be essential for direct interactions with the relative membrane-localized chaperones. Rather surprisingly, no mutation or condition has been shown to block PM transporters in the Golgi or in other post-ER compartments (secretory vesicles or endosomes). It seems that specific, de novo transporters, such as the CFTR channel involved in cystic fibrosis or the UapA purine transporter, exit the ER in COPII vesicles, but subsequent sorting steps involve an unconventional mechanism of trafficking that basically requires clathrin heavy chain and proper actin organization [85–87].

## 4. Tails control transporter ubiquitination, endocytosis, turnover or recycling

All fungal transporters and several mammalian transporters that have been studied are downregulated in response to physiological or stress signals, or in response to transport activity in the presence of a continuous supply of substrates [88–90]. This turnover control takes place at the level of the PM, leading to internalization of the transporter by endocytosis. Internalized endocytic vesicles are sorted in early endosomes, which mature to late endosomes/multivesicular bodies and eventually fuse with the vacuole/lysosome, the site of their degradation. Some transporters, after endocytosis, can be recycled back to the PM via sorting endosomes or specialized transporter vesicles [91–94]. A common theme in all cases of transporter endocytosis, and subsequent vacuolar degradation or recycling, is ubiquitination and de-ubiquitination of specific Lys residues [95,96]. Lys residues

**Table 1.** Paradigms of N- or C-terminal cytosolic motifs and other non-conserved functional sequences in transporters. Acidic motifs marked with superscript 1 control α-arrestin-dependent, HECT-type ubiquitination, followed by endocytosis, in distinct transporter families, and probably define the target sequence of specific α-arrestin adaptors. Notice that distinct acidic motifs (e.g. in Gap1) are crucial in ER exit, rather than endocytosis. The motifs marked with superscript 2 are very probably serving the same unknown function in plants and animals. Marked in bold are nearly absolutely conserved residues in the families described.

| motif | tail | distance (aa) from neighbouring TMS | taxon | role | experimental work |
|---|---|---|---|---|---|
| *NCS1 (TMS12)* | | | | | |
| **N**-X-**D**/S-Φ-X-**P** | N | 10 from TMS1 | ubiquitous (prokaryotes, fungi, plants) | substrate specificity/interaction with distal part (20 aa) of C-tail and internal loops | FurE |
| acidic motif **E/D**-X-**E**-**E**[1] | C | 3 from TMS12 | fungi | ubiquitination/endocytosis | FurE, FurD, FurA |
| pair of Ks in the last 10−20 residues | | 20−30 from TMS12 | | | |
| *NAT (TMS14)* | | | | | |
| **G**-**D**-**Y**-**D**-**Y** | N | 27−40 from TMS1 | fungi | ER exit, oligomerization/lipid interactions? | UapA |
| acidic motif | C | 1 from TMS14 | fungi | ubiquitination/endocytosis | UapA |
| **E**/D-**E/D**-X-**E**/D-**E/D**[1] | | 2 from the C-end | | | |
| *K572* | | | | | |
| **D/E-Q**-L/Φ-X-X-Φ-X-**Y**-**C**-**I**-X-**S** | N | next to TMS1 | plants | ? | — |
| **D/E-R-G**-X-X-**W**[2] | C | 5−8 from TMS14 | plants | ? | — |
| **D/E**-X-**R/A**-X-X-**E-F-Y**-X-**L-P**-X6-**F** | C | most distal C-segment | plants | ? | — |
| **E/Q-R-G**-Φ-X-X-**W**[2] | C | 5 from TMS14 | metazoa | ? | — |
| **Y**-D/N-X-**P-F-G**-Φ | C | 27−37 from TMS14 | metazoa | ? | — |
| Φ-Φ-**P**-Φ/F-X-**P** | C | 48−59 from TMS14 | metazoa | ? | — |
| *APC (TMS12)* | | | | | |
| K8, K16 | N | 75−82 from TMS1 | yeast | Bul- or Aly-dependent ubiquitination/endocytosis | Gap1 |
| distinct patches of non-conserved residues (e.g. P-I-T-I or G-N-H) | | 56−72 from TMS1 | | | |
| distinct patches of non-conserved residues (e.g. V-D-L-D or E-I-A-E) | C | 12−32 from TMS12 | yeast | Bul- or Aly-dependent ubiquitination/endocytosis | Gap1 |
| acidic motif **D/E**-X-**D/E** | C | next to TMS12 | yeast | COPII interacting motifs | Gap1 |
| K46, K48 | N | 52−54 from TMS1 | yeast | ubiquitination/endocytosis | Can1 |
| acidic motif **D-E-D-E**-G-**D**[1] | | 10 from TMS1 | | | |
| E-L-K | | next to TMS1 | | | |

(Continued.)

**Table 1.** (Continued.)

| motif | tail | distance (aa) from neighbouring TMS | taxon | role | experimental work |
|---|---|---|---|---|---|
| K27, K28 | N | 30–31 from TMS1 | yeast | ubiquitination/endocytosis | Mup1 |
| acidic patch | | 5 from TMS1 | | | |
| (X₂-**D**-X₃-**G**-X-**Q**-**F**-X-**T**-X-**L**-X)[1] | | | | | |
| *NSS (TMS12)* | | | | | |
| A-I-**Y**-**K**-X4-P-**G**-X-**Φ**-**X**-**D**/E/**Q**-**K**/**R**-X7-**P** | C | just from TMS12 | chordate | transport activity? | — |
| | | | NSS | stability? | |
| **P**-**K**-**E**-**V**-**E**-**L**-I-**L**-**V**-**K**-**E**-**Q**/**H**-**N**-**G**-**V**-**Q**-**F**/**V**-**T** | N | 16 from TMS1 | vertebrate | specificity? | — |
| | | | DAT | trafficking? | |
| **L**-**Φ**-X-X-**G**-X-**V**-**R**-**Q**-**F**-X-**L**-X-**W**-**W**-**L**-**K**-**V** | C | most distal C-segment | vertebrate | specificity? | hDAT |
| (PDZ-binding motif L-K-V) | | | DAT | stability/recycling | |
| *MFS (TMS12)* | | | | | |
| **F**-**Q**-**I** | N | 13 from TMS1 | mammals | endocytosis/recycling? | GLUT4 |
| **L**-**L** (or R-T-P-**S**-**L**-**L**-E-**Q**) | C | 18 from TMS12 | mammals | endocytosis/recycling? | GLUT4 |
| **T**-**E**-**L**-**E**-**Y**-**L**-**G**-**P** | C | 7 from the L-L motif | mammals | recycling | GLUT4 |
| (or **L**-X-X-**L**-X-**P**-**D**-**E**-X-**D**) | | (most distal C-segment) | | | |
| **D**-**R**-**S**-**G**-**K**-D-**G**-**V**-M-**E**-**M**-**N** | C | 20 from TMS12 | mammals | targeting | GLUT3 GLUT5 |
| polyproline, phosphorylation consensus | C | most distal C-segment | mammals | recycling | VGLUT1 |
| | | | | | VGLUT2 |
| *several transporter families* | | | | | |
| **FF**, **YY**, **LL**, **FY**, **Φ**X**Φ**X**Φ** | N/C | variable positions | eukaryotes | endocytosis/recycling/COPII interacting motifs | several transporters |

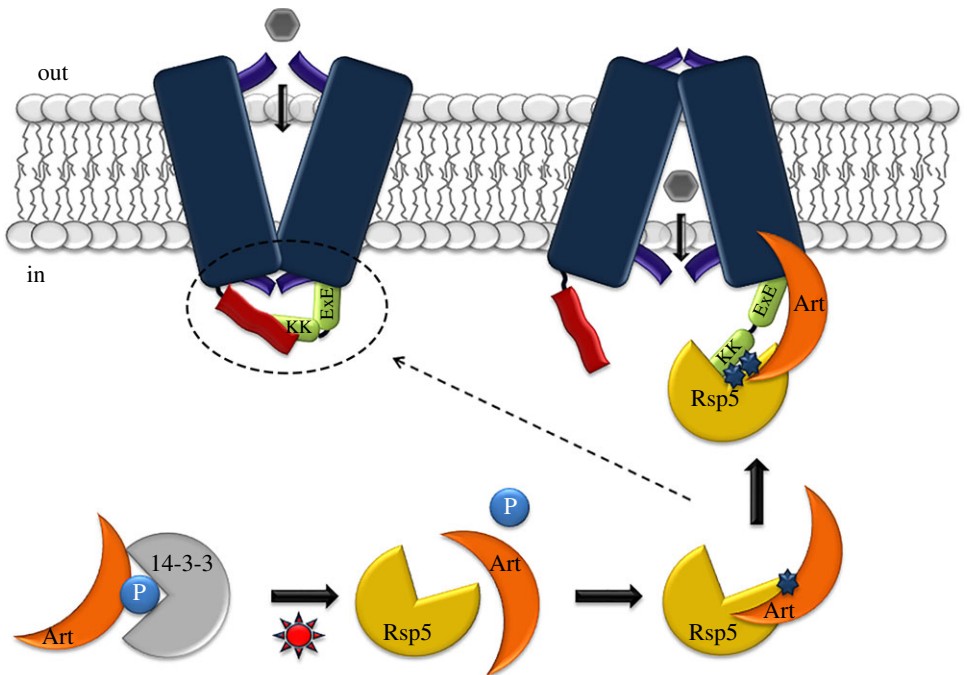

**Figure 2.** Generalized model of transporter endocytosis highlighting the crucial role of N- and C-tails, based mostly on data concerning the FurE purine transporter, but also integrating critical findings from studies with Gap1 and Jen1 transporters (see text). The figure shows that in the outward-facing conformation the N- and C-tails are in close contact with each other and with other cytoplasmic domains of the transporter (e.g. the inner gate shown in purple), while in the inward-facing conformation the N- and C-tails separate to become more relaxed to recruit cytoplasmic effectors, such as those leading to ubiquitination and endocytosis. The figure includes a HECT-type ubiquitin ligase (Rsp5), its α-arrestin adaptor (Art) and a 14-3-3 protein that inhibits Art association with Rsp5 via phosphorylation. Upon a signal eliciting endocytosis (depicted by a blue-red star), Art is de-phosphorylated, acquires high affinity for Rsp5, which ubiquitylates Art, and the Art–Rsp5 complex is recruited to transporter tails. The relaxed topology of the cytosolic tails in the inward conformation permits more efficient recruitment of the Art–Rsp5 complex than the 'hidden' tails in the outward-facing conformation. The specific lysine residues (KK) and acidic motifs (e.g. EXE) necessary for ubiquitination are located either in the C-terminus (as in the figure) or in the N-terminus, but the interaction of termini is crucial for accessing these elements, so that both termini are critical for endocytosis.

modified by ubiquitination are present nearly exclusively in transporter tails [89,97]. Transporter ubiquitination is carried out by HECT-type ubiquitin ligases of the Nedd4/Rsp5 type [97–100], recruited to transporter tails by adaptor proteins called α-arrestins [101–105].

Importantly, increased ubiquitination and endocytic turnover of transporters also depend on the conformational dynamics of transporters themselves. Thus, several transporters are much more vulnerable to ubiquitination and endocytosis when they actively transport their substrates, a phenomenon known as activity-dependent or substrate-dependent endocytosis [40,89,106]. Genetic and biochemical evidence from a number of studies with fungal transporters supports that particular conformations of the transporter, triggered after substrate binding (e.g. substrate-occluded or inward-facing topology acquired after substrate release), increase transporter accessibility to arrestin adaptors [40,89,107]. As will be shown later, transport tails are important for conformation-dependent endocytosis.

In the purine-related UapA and FurE transporters of *A. nidulans* a short acidic motif (**E/D**-X-**E-E**; see table 1) located at the cytosolic C-terminus was shown to be essential for ubiquitination and/or endocytosis [17,39]. In both UapA and FurE, this acidic motif is located immediately next to the last TMS and 23–25 residues upstream from the Lys residues modified by ubiquitination. Arrestin sorting signals were also recently described in the N-terminus of the methionine or arginine transporters Mup1 and Can1 of *S. cerevisiae* (table 1). Similarly to the *A. nidulans* examples, these motifs include an extended acidic patch in close proximity to

TMS1 and are close to the ubiquitinated lysines [104]. These findings suggest a common mechanism for recognition of transporters by arrestins in ascomycetes, based on opposing charge interactions of the arrestin C-terminal basic region and the acidic patches present in either the N- or C-terminal cytosolic regions of evolutionary and functionally distinct transporters. However, although this interaction is necessary for transporter ubiquitination and endocytosis, it might be insufficient, as it is usually context dependent. In line with this, in the case of Can1, Ala substitutions in the acidic patch reduce, but do not abolish ubiquitination and endocytosis, and, in addition, efficient arginine-elicited endocytic turnover seems to require a shift to the inward-facing conformation, which modifies the positioning of a distinct specific tripeptide (**E-L-K**), just upstream from TMS1 (table 1) [107]. Interestingly, in the Gap1 transporter of *S. cerevisiae*, both the N- and C-termini contain elements (table 1) necessary for ubiquitination and endocytosis by distinct arrestins [108], in response to different signals, but the Lys residues shown to be ubiquitinated are present solely in the N-tail [109,110]. This is in line with some other cases, where mutations blocking ubiquitination and endocytotic turnover are located not only in the terminal segment that includes the ubiquitination-specific Lys residues and a nearby arrestin target, but also in the *opposite* cytosolic terminus. This is nicely exemplified in studies concerning the Fur purine-related transporters of *A. nidulans* ([16,17]; figure 2). In this case, relatively short specific truncations or triple Ala substitutions in either terminus of the FurE transporter block its turnover. The C-terminus of FurE contains the Lys residues

royalsocietypublishing.org/journal/rsob    Open Biol. 9: 190083

and the arrestin acidic target sequence necessary for ubiquitination, but the N-terminus has no obvious sequence that could be predicted *a priori* to participate in endocytic turnover. Independent *in vivo* bifluorescence complementation (BiFC) assays have shown that the N- and C-termini of FurE come into close contact during transport of substrates.

One extensively studied case of transporter recycling is the insulin-stimulated glucose transporter GLUT4 in mammalian cells. GLUT4 traffics through several distinct intracellular compartments, including early endosomes, intermediate transport vesicles, recycling endosomes and/ or the TGN, and insulin-responsive vesicles (IRVs) (also called GSVs for glucose transporter storage vesicles). Initially, two distinct motifs have been identified, a phenylalanine-based motif (**F-Q-Q-I**) in the N-terminus and a di-leucine motif (**L-L** or R-T-P-S-**L-L**-E-Q) in the C-terminus, which function autonomously as internalization motifs, and so presumably interact with AP-2 at the PM [111,112]. Additional terminal motifs seem to regulate the subcellular trafficking, recycling or targeting of GLUT4. For example, the sequence **T-E-L-E-Y-L-G-P** (or **L-X-X-L-X-P-D-E-X-D)**, 8 residues downstream from the L-L motif in the C-terminus, is critical for sorting, via the endosome, to the TGN [113]. C-terminal motifs seem also to play an important role in respect to the final targeting of GLUT transporters. The C-terminal sequence **D-R-S-G-K-D-G-V-M-E-M-N** is an example of autonomous apical targeting [114].

The vesicular glutamate transporters 1 and 2 (VGLUT1 and VGLUT2) of the $Na^+/P_i$ co-transporter family, expressed in the neuron-rich regions of the brain, also show a differential functional dependence on their cytosolic termini owing to variable motifs. VGLUT1 contains two polyproline domains that interact with the endocytic protein endophilin for recruiting VGLUT1 to a fast recycling pathway. Additionally, both VGLUTs contain multiple di-leucine similar trafficking motifs that direct trafficking by distinct pathways that use different clathrin adaptor proteins [115–117].

Reuptake of synaptically released neurotransmitters by NSS transporters is the primary mechanism to control duration and strength of neurotransmission. In most cases, differential internalization rates and post-endocytic sorting of NSS transporters are controlled by different elements in their termini. For example, the norepinephrine (NET) and DAT transporters have distinct trafficking properties, and exchange of domains revealed that this difference was determined by non-conserved structural elements in the N-terminus, while the C-terminus had no effect [118]. In table 1 we describe an N-terminal conserved motif (P-K-E-V-E-L-I-L-V-K-E-Q/H-N-G-V-Q-F/Y-T) that might be critical for DAT trafficking properties. DAT also encodes a C-terminal PDZ-binding motif (**L-K-V**), which seems to be important for DAT PM stability, but also for exit from the retromer complex and recycling back to the PM [119–121]. Similar to DAT, SERT is mainly sorted to late endosomes and lysosomes, and if recycled (an issue under strong debate) seems to follow the Rab4 recycling pathway [122]. In contrast to the importance of the C-tail of DAT in PM stability recycling, mutations in the C-terminal region of SERT affect ER exit by either abrogating putative SEC24-binding motifs (P-G or R-I-I) or by impairing folding of the transporter. The R-I-I motif of SERT might be homologous to the sequence R-L found to be critical for ER exit of other NSS transporters [123]. Differential targeting of NET and DAT to the basolateral and apical membranes has also been found to be due to differences in their N-termini [124]. An apical localization signal for GAT1 (GABA transporter 1) also appeared to reside in the N-terminus, although a basolateral localization signal for GAT2 was located to the C-terminus [125].

## 5. Tails allosterically affect transport activity and substrate specificity

One of the most interesting novel concepts concerning transporter tails is their role in regulating transport function and substrate selection, apparently via complex allosteric interactions with each other and cytosolic loops of transporters. Emerging evidence supports that the N- and/or C-tails of LeuT-like transporters are crucial in the rocking-bundle mechanism of transport [47,126,127].

In the outward conformation of LeuT and DAT an important salt bridge is formed between the N-terminal tail and TMS8, which is well conserved in the NSS family [128]. It concerns the pairs Arg5–Asp369 in LeuT and Arg27–Asp435 in dDAT. It has been suggested that this interaction forms the cytoplasmic inner thin gate of the transporter. Studies on the human dopamine transporter (hDAT), further suggested that there is a network of interactions, including the salt bridge Arg60$^{hDAT}$–Asp436$^{hDAT}$ and interactions with the cytoplasmic end of TMS6, stabilized by a cation-π interaction between Arg60$^{hDAT}$ and Tyr335$^{hDAT}$, and with Val259 (L4) and Glu428 (TMS8) [129,130]. Indeed, in the crystal structure of *Drosophila* dDAT (Protein Data Bank (PDB) 4m48) residue Tyr334$^{dDAT}$ (TMS6) forms hydrogen bonds with five other residues, namely Glu427 and Thr431 in TMS8, and Arg27, Glu28 and Thr29 in the N-tail. Additionally, two other salt bridges are formed between the N-tail and L2, namely Glu26–Arg92 and Asp25–Lys93 (figure 3*b*). These interactions reveal the possible role of the N-terminal cytosolic segment as a barrier between the substrate and the intracellular environment [131]. In the recent crystal structure of SERT the corresponding inner thin gate is also formed as a salt bridge between Arg79 and Asp452 [132]. Another salt bridge between the C-terminal Glu615 and Arg152, belonging in the cytosolic L2 loop, has been proposed to be crucial for the folding and probably trafficking of SERT, but this was not confirmed in the crystal structure of the C-terminus [133–136]. Overall, it appears that disruption of such networks of interactions might initiate the conformational conversion of NSS transporters from outward to inward topologies, regulating access and facilitating permeation through propagation of motions. Furthermore, as has been mentioned above, the N-terminal domains of the mammalian neurotransmitter transporter are longer than the prokaryotic, probably enabling other functions except gating. Molecular dynamics (MD) studies on hDAT have indicated that the distal segment of the N-terminus forms coulombic interactions with negatively charged lipids of the membrane bilayer, sustaining transport specifically in eukaryotic environments [50,137]. Apparently, evolution has added further regulatory roles at both the N- and C-termini of eukaryotic transporters [138].

A crucial role in the mechanism of transport of the C-tail segment was found in members of the APC family, namely the AdiC and GadC amino acid antiporters, which conform to the LeuT fold [27]. AdiC (arginine–agmatine exchanger)

royalsocietypublishing.org/journal/rsob   Open Biol. 9: 190083

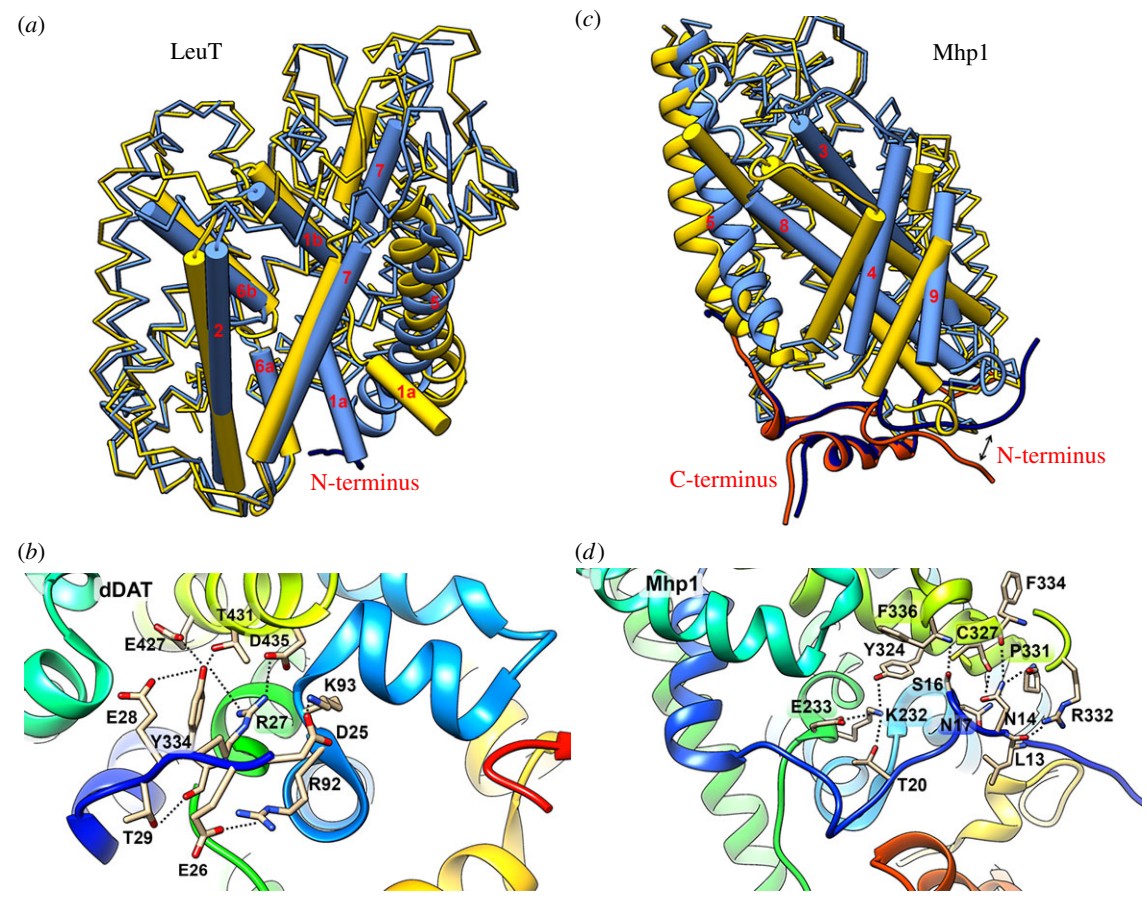

**Figure 3.** Structural insights into LeuT-fold transporters with emphasis on the functional role of cytosolic terminal domains. (*a*) LeuT dynamic topology depicting the tilt of the 'bundle' domain (TMS 1,2,6,7) from the outward (blue) to the inward (gold) conformation. Notice also the displacement of TMS1a and TMS5 (PDB entries 3tt1, 3tt3). (*b*) Interactions of the cytosolic N-tail (deep blue) with internal loops in dDAT (PDB entry 4m48). (*c*) Mhp1 dynamic topology showing the rocking movement of the 'hash' domain (TMS 3,4,8,9) from the outward (gold) to the inward (blue) conformation. Notice also the displacement of TMS5 (PDB entries 2jln, 2x79). (*d*) Interactions of the cytosolic N-tail (deep blue) with internal loops and TMS6 and TMS8 in Mhp1 (PDB entry 2jln). All snapshots were generated by Chimera 1.10.

exists as a homodimer with TMS11 and TMS12 mediating the interaction between protomers. GadC (glutamine–GABA antiporter) shares sequence homology with AdiC and adopts the same fold [139]. AdiC has been crystalized in the outward and outward-occluded conformations, while GadC has been crystallized in an inward-open conformation [28,140]. Comparison between these two structures showed that switching from outward to inward is correlated with a structural rearrangement of the 'bundle' consisting of TMSs 1, 2, 6 and 7, similar to LeuT, while the 'hash' TMSs 3, 8 and 9 exhibit smaller conformational changes. Interestingly, in the inward topology of GadC the C-terminal is folded within the open binding cavity, thus blocking the substrate-binding site. Substrate transport should necessarily include C-terminal displacement, which has been correlated with the multiple interactions formed by several basic residues. The C-terminal of AdiC is shorter than GadC and notably, in the outward-facing conformation of the transporter, seems to interact with the cytosolic domains of the opposite protomer of the dimer in the crystal [28,141,142]. Apart from the C-terminal interactions found in the AdiC/GadC structures, the N-terminal regions of APC transporters do not seem to form any of the interactions described for the NSS family. Only a weak interaction between His8 and Ala320 in L8 can be encountered, while no salt bridge or hydrogen bond network is observed.

In the case of the benzyl-hydantoin–sodium symporter Mhp1 three structures, representing the conformational changes from outward to outward-occluded and to inward, have been crystallized. Based on these, Cameron, Henderson and co-workers [7,31,143,144] have proposed a somehow different model of transporter conformational change during ligand translocation, compared with LeuT and NSS. It appears that, in Mhp1, after ion and substrate binding in the outward-open structure TMS10 bends over the cavity in the occluded state, thus triggering the rotation of 30° of the hash motif relative to the bundle. Then TMS5 flexes in order to open the cavity to the intracellular side (figure 3*c*; [7,143,144]). The N-terminal region of Mhp1 expands over the cytoplasmic surface interacting with L2, L6, L8 and L10 loops, as well as with the C-terminal region. The shape of the N-terminal fragment seems to remain stable from the out-ward- to the inward-facing structure curved in the vicinity of Pro15. Interactions between the N-terminus and L8 are encountered in the inward-facing structure, namely Asn14–Phe334 (backbone), Arg332–Leu13 (backbone), Asn17–Cys327 (backbone), Asn14–Pro331 (backbone) and Ser16–Phe336 (backbone) (figure 3*d*); however no salt bridge, similar to that shown in NSS members, is formed. Interestingly, another network of hydrogen bonds is also formed between Thr20–Glu233(TMS6)–Lys232(TMS6)–Tyr324 (TMS8), linking the N-terminus, TMS6 and TMS8, as was observed with DAT.

All the above interactions are disrupted in the inward-facing conformation following the rearrangement of the hash motif of TMS3, 4, 8 and 9, although the N-terminal fragment is shown to be quite rigid with a relatively small-scale reorientation of the segment between Pro15 and Glu8. This suggests that although important differences are observed between Mhp1 and LeuT in the conformational changes related to the mechanism of substrate translocation it appears that the interruption of the contacts between the N-terminus and L8 in the inward-facing structure is similar in both cases, supporting the concept of the thin inner gate.

# 6. Tails affect fine gating and substrate specificity in NCS1 transporters

In the course of analysing the roles of the cytosolic C- and N-termini of members of the Fur group of the NCS1 family in *A. nidulans* it became apparent that terminal segments host elements necessary not only for endocytic turnover and subcellular sorting, but surprisingly also for substrate specificity [16,17]. In particular, the 38-amino-acid-long N-tail of FurE, an allantoin–uric acid–uracil transporter, has been shown to include distinct segments critical for endocytosis (residues 1–11), substrate specificity (residues 12–29) and ER exit (residues 30–32, 36–38). The C-tail also contains elements important for endocytic turnover and substrate specificity, the first corresponding to a di-acidic motif Glu–Glu, 11 residues upstream from the end of the protein, and the second concerning the last 29 residues of the transporter. The specificity changes obtained were of two kinds. Either they enlarged the set of substrates transported by FurE to include xanthine, in addition to uracil, allantoin and uric acid, or, instead, they restricted specificity to uracil and allantoin. Mutations enlarging specificity concerned Ala substitutions in the N-tail conserved motif **N**-X-**D**-Φ-**D**-**P** (residues 24–29). The ones that restricted specificity were deletions either of the most distal part of the N-tail (residues 1–21) or of the C-tail (last 29 residues), as well as Ala substitutions of N-tail residues 15–17.

The similarity of phenotypes obtained by mutations in the N- and C-termini (i.e. block in endocytosis and changes in specificity) was suggestive of an involvement in a common mechanism controlling FurE function. Indeed, intramolecular BiFC analysis showed that the FurE cytoplasmic N- and C-termini interact dynamically, and this interaction is transport activity dependent. It was thus proposed that in the absence of substrates FurE is found in a rather stable outward-facing conformation, which brings the cytoplasmic N- and C-termini into close contact, and this permits the reconstitution and detection of a fluorescence signal. In the presence of substrates, the transporter is 'forced' to continuously alternate from the outward- to inward-facing conformation, and thus the termini become dynamic and do not reconstitute a stable fluorescence signal [16]. The interaction of the N- and C-termini is not essential for transport activity, as FurE versions truncated in either one or both termini can still function, but show modified specificities. Independent genetic evidence supported that the interaction of the FurE termini is critical for the opening and closing of the substrate translocation pathway, and thus controls the gating process. This evidence concerns the finding that genetic suppressors restoring specificity changes (i.e. loss of uric acid transport)

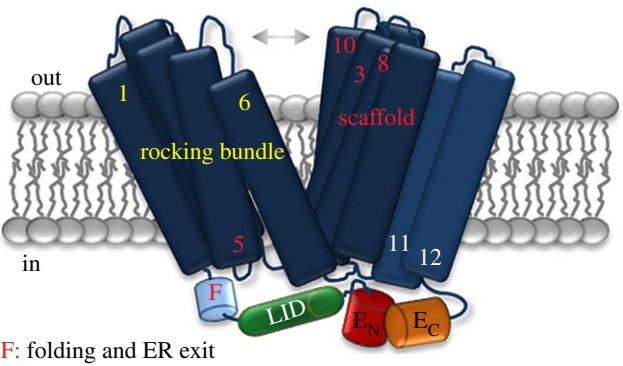

F: folding and ER exit
$E_N$ and $E_C$: N- and C-tail endocytic turnover

**Figure 4.** The multiple roles of both termini of FurE in folding, ER exit, endocytosis and substrate specificity. The figure is based mostly on data concerning the FurE transporter, which contains a LeuT-like fold. Details are described in the text and in [17]. LID stands for the central part of the N-tail that is specifically involved in interactions with several other cytosolic internal loops and thus allosterically regulates the functioning of the outer and inner gates and substrate specificity. $E_N$ and $E_C$ are the distal parts of the tails that interact dynamically with each other during transport, and thus recruit ubiquitination and endocytosis factors, but also the positioning of the LID, which, in turn, affects the function and specificity of the transporter. F is the part very proximal to TMS1 that is critical for the correct folding of TMS1 and of the transporter, and thus affects packaging to COPII vesicles and ER exit.

of N- or C-terminal mutants are located at a tentative external gate, along the substrate translocation path, or in flexible loops that act as dynamic hinges during transport catalysis [16,17]. Additionally, MD and further mutational analysis supported that specific polar residues of the N-terminus (Asn24, Asp26 and Asp28) interact dynamically with residues of several internal cytosolic loops of FurE, and this interaction controls the opening and closing of the outer and inner gates [17]. Given that the positioning of the N-terminus also depends on its interaction with the C-terminus, both termini are thus involved in a dynamic molecular cross-talk with the internal loops of FurE (figure 4).

# 7. Tails in transporter–lipid interactions and oligomerization

It is now well established that many transporters form dimers or oligomers that are important for effective trafficking to the membrane, function and/or regulation of function [13,14,145]. A well-studied case of transporter dimerization is that of UapA, where evidence based on dominant negative mutations, light scattering, pull-downs, BiFC assays, and eventually crystallization confirmed that dimerization occurs *in vivo* and is required for function and specificity [13,14,38,40]. In addition to UapA and its structural homologues [146–151], other transporters, mostly those of the LeuT or 5+5 fold, also seem to form oligomers. In several of these cases, approaches such as fluorescence resonance energy transfer (FRET) [152], cross-linking [153], pull-downs [154], co-immunoprecipitation [155] and size-exclusion chromatography–multiangle light scattering (SEC–MALS) [156] have been employed to prove functional oligomerization [13,14,145]. Evidence also supports that some members of the NSS transporter family are dimers, while others are monomers, and still others can be oligomeric

depending on their localization [157–161]. Other transporters that oligomerize are members of the SWEET transporter family, the ammonium transporter/MEP/Rh transporter family and the MFS ([13,14] and references therein). In all the above cases the role of membrane lipids in the formation of oligomers is emerging as a critical issue. Two types of interactions of membrane proteins and lipids are recognized. The so-called annular membrane lipids form a ring around individual transporter units and thus stabilize them via H-bonding with lipid head groups and hydrophobic interactions. Additionally, membrane lipids might interact with high affinity with specific residues present deep in the transporter body, often in the interface of dimers. Such interactions regulate membrane protein structure and function [13,14,145,162].

Are transporter tails critical for oligomerization or is this information encoded solely in the main transmembrane body of transporters? Recently, mass spectrometry was used to relate the presence of interfacial lipids and oligomeric stability, and discover how lipids act as key regulators of membrane protein association [163]. This approach showed the absence of interfacial lipids in the mass spectra of membrane proteins with high oligomeric stability. However, in proteins with the lowest oligomeric stability, lipids were present within the dimer interface, and delipidation or mutation of lipid binding sites abrogated dimer formation. Interestingly, in the NapA $Na^+/H^+$ antiporters from *Thermus thermophilus*, an additional N-terminal helix, not present in the orthologous *Escherichia coli* NhaA transporter, strengthens interface interactions, removing the requirement for lipids to stabilize dimer formation. This N-terminal helix seems to interact with a subset of annular lipids as further structural support to facilitate large-scale conformational changes within the membrane [164]. The trimeric betaine transporter BetP is also a case where interactions of tails with lipids seem critical for function. A dynamic interaction between the C- and N-terminal domains of adjacent protomers has been shown to modulate transport activation. In brief, it has been proposed that the C-terminal domain changes its interaction with the N-terminal domain of its own promoter and negatively charged lipids to an interaction with the N-terminal domain of an adjacent protomer and lipids bound to the central cavity of the BetP trimer [165,166]. Another paradigm of transporter tails involved in functional interactions with PM lipids is that of the human dopamine transporter hDAT, which contains long intracellular N- and C-terminal domains that seem to be implicated in the transporter function, and possibly oligomerization, *in vivo* [48,167]. Specifically, its N-terminus controls the efflux of the substrate through hDAT, although it is not critical for dopamine import *per se*. A computational model of the N-terminus of hDAT obtained by an *ab initio* structure prediction, in combination with MD simulations in the context of a lipid bilayer, revealed that the N-tail is a highly dynamic domain, but also contains secondary structure elements that remain stable in the long MD trajectories of interactions with the bilayer. These stable elements include specific residues that interact with charged PIP2 (phosphatidylinositol 4,5-biphosphate) or PS (phosphatidylserine) lipids and thus seems to control the function, and possibly the stability, of the DAT oligomers.

Still other reports have shown that lipids other than phospholipids in particular sterols seem to play important roles in the functioning of DAT and other NSS transporters [168–170]. Recent crystal structures of DAT revealed the presence of two conserved cholesterol-like molecules bound to it. Relative MD simulations suggested that outward-facing DAT in the absence of cholesterol undergoes a conformational modification that mimics the transition to an inward-facing conformation. In the presence of bound cholesterol these conformational changes are inhibited by immobilization of the intracellular interface of TMS1a and TMS5. Coarse grain MD simulations further suggested that cholesterol binding sites in DAT are conserved in other NSS transporters. Cholesterol binding and control of transport function have also been related to several other transporters, but in most cases the proposed cholesterol binding sites (known as CARC–CRAC sequences) are located in membrane-embedded segments, rather than in the tails of transporters [171,172]. This however does not exclude that tails might also be functionally involved in the binding of sterols or other lipids (e.g. sphingolipids).

## 8. Tails of transceptors in signalling

Transceptors are functional transporters that possess both solute transport and direct receptor-like signalling activities [173,174]. These proteins sense nutrients by ligand-induced conformational alterations recognized by downstream intracellular effectors. Best-studied transceptors include yeast transporters specific for amino acids (Gap1), phosphate (Pho84), ammonium (Mep2), sulfate (Sul1 and Sul2), iron (Ftr1) and zinc (Zrt1) [175–179]. All these high-affinity transporters signal to cAMP-independent activation of the protein kinase A (PKA) pathway. Transceptors are usually highly induced upon relative nutrient starvation and rapidly down-regulated by substrate-induced endocytosis. Transceptors for nitrate, ammonium, sulfate or nucleosides have also been identified, at least tentatively, in filamentous fungi (*Neurospora crassa* and *Ustilago maydis*), plants (*Arabidopsis thaliana*), protozoa (*Leishmania mexicana*) and in human cells [180]. The concept of transceptors was based on evidence showing that transport and signalling can be genetically uncoupled, that is, maintenance of signalling in the absence of transport or vice versa, or by the identification of competitive and non-competitive inhibitors of transport that promote signalling.

What distinguishes a transporter that acts solely in transport from a transporter that also acts as a signal receptor? Are the cytosolic termini of transceptors, which in some cases are particularly long (e.g. Sul1 and Sul2; [178]), critical for signalling, as it is in bona fide receptors? Some evidence exists that indeed the termini of transceptors are important for signalling in addition to transport catalysis and turnover. In fact, transport and signalling are interrelated and both are dependent on substrate/ligand-elicited conformational changes. It seems that different substrates or ligands bind and elicit distinct substrate/ligand-specific conformational changes in transceptors, thus promoting distinct interactions of termini with downstream effectors. Although immediate downstream transceptor-interacting proteins have not been identified, there is evidence that cytosolic termini are important for transport, and thus possibly for signalling too.

A well-studied case concerning the role of termini in transceptors is that of the yeast Mep2 ammonium transporters [177,181]. Under nitrogen-sufficient conditions, non-phosphorylated Mep2 exhibits shifts in cytoplasmic

loops and the C-terminus that occlude the cytoplasmic exit of ammonium. Under nitrogen-depleting conditions, phosphorylation of Ser457 within the C-terminal region causes Mep2 opening, via eliciting modifications of interactions of the C-terminus with internal loops L1 and L3 (Mep2/Amt-1 proteins are trimeric assemblies in which each monomer consists of 11 TMSs, an extracellular N-terminus and an intracellular C-terminus). Additionally, phospho-mimicking Mep2 versions also show large conformational changes of a region of the C-terminus. Interestingly, the *A. thaliana* homologous ammonium transporter (Amt-1) is allosterically regulated by phosphorylation, but in an opposite way [182]. Findings with Mep2 and Amt-1 have strongly supported that the C-terminus mediates allosteric regulation of ammonium transport activity, via phosphorylation. More specifically, it has been proposed that close intra-monomeric C-terminus interactions with L1 and L3 generate open transceptors, whereas weakening of these interactions leads to inactivation of transport. The crystal structure of a homologous prokaryotic (*Archaeoglobus fulgidus*) ammonium transporter further suggests that the C-terminus interacts physically, in a phosphorylation-dependent manner, with cytosolic loops of neighbouring subunits [183]. Such conformational coupling between monomers might provide tight regulation not only for the transporter but also for sensing nutrients. These studies, however, have not addressed the role of C-terminus interactions in signalling.

Truncation of the last 14 or 26 amino acids from the C-terminus creates a constitutively active allele of the Gap1 transceptor that resides permanently in the PM and causes constitutive over-activation of the PKA pathway, provided it is expressed in a mutant strain that allows such truncated alleles to be secreted to the PM. However, the specific Gap1 conformation that elicits signalling remains unknown [184,185]. Further work is needed however to determine the specific Gap1 conformation which elicits signalling, and to elucidate the molecular mechanisms through which this particular Gap1 conformation is sensed and the signal transduced to downstream targets.

# 9. Conclusion

Current findings are consistent with the idea that the size of eukaryotic transporter termini increased during evolution, providing additional flexible elements that enabled the generation of novel transporter functions and specificities, as well as specific mechanisms for controlled trafficking, turnover and/or oligomerization elements not shared with their bacterial homologues. Distinct but interacting roles for distal and proximal segments of the larger N- and C-termini in folding, turnover, ER exit and trafficking to the PM, or in transport function *per se*, have been identified in several eukaryotic transporters. One of the most critical emerging issues is the interaction of tails with membrane lipids and other cytosolic domains of the transporter main body. The case of the FurE transporter serves as an instructive paradigm on how specific transporter tail interactions with internal loops, and most probably with phospholipids, can allosterically finely regulate the gating process, and thus substrate specificity, while distinct tail interactions can also regulate folding, trafficking and turnover. Evolution of functional tails may represent a naturally selected adaptation required for the integration of new functions in eukaryotic transporters, which face increasingly complex cellular challenges of regulation in response to differentiation and metabolic needs.

Data accessibility. This article has no additional data.

Competing interests. We declare we have no competing interests.

Funding. This work was supported by a 'Stavros S. Niarchos Foundation' grant to G.D and E.M.

Acknowledgements. We are grateful to Georgia Papadaki for help in figure design.

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
