## [Reviewer comments · Open Biology]

Review History

RSOB-19-0083.R0 (Original submission)

Review form: Reviewer 1

Recommendation

Accept with minor revision (please list in comments)

Are each of the following suitable for general readers?

- a) **Title**
Yes
- b) **Summary**
Yes
- c) **Introduction**
No

Is the length of the paper justified?

No

Should the paper be seen by a specialist statistical reviewer?

No

Is it clear how to make all supporting data available?

Not Applicable

Is the supplementary material necessary; and if so is it adequate and clear?

Not Applicable

Do you have any ethical concerns with this paper?

No

Comments to the Author

Diallinas and Mikros

Tales of tails in transporters

The manuscript reviews the state of the art with respect to the role of N and C-terminal regions in terms of transporter structure and function. These roles are emerging and the current paper, if edited somewhat, will be a useful addition to the current literature in this area. The figures are good quality and help to highlight key issues covered in the manuscript. My major issue with the manuscript is that it is far too long and relatively unfocused. There are lengthy descriptions of aspects of transporter biology that are not directly relevant to the key subject of the manuscript that distract from the topic being discussed. A substantial reduction in the length of the manuscript will make this easier to read and be more in keeping with the title of the paper.

In addition, the manuscript could do with an English edit.

Comments:

Page 2 Emphasis should be made at the start of the manuscript that the focus is secondary active transporters. A single mention of ABC transporters is made further down the page and this hardly seems relevant in this manuscript.

Page 2 Para 2 Line 22 Should really cite Forrest et al, 2011 here also.

Pages 3-4 Include a lengthy description of the different families of transporters. This could be cut down to just focus on the differences in the tail regions.

Pg 5 5th from last line. The text states that a specific motif might be critical for oligomerisation of UapA. What is the evidence for this?

Page 9-13 I think this could either be cut down or additional sections added

Page 13-14 Could dramatically cut down on the second paragraph in section 5. Only critical sentence is last one "Emerging evidence supports that the N- and ... " MD is referred to throughout the manuscript as "showing" things. I would prefer the word "indicate"

Pg 16-17 Again another very long section. Should make more concise to focus on the points relevant to function of the tail region.

Pg 18-19 The section on oligomerisation has a very long introduction-could be summarised in a couple of sentences citing the Alguel and Cecchetti reviews.

Pg 19-20 The relevance of lipid interactions should be only the tail regions.

Pg 20 Last paragraph is very vague-should be really factual focusing on known material.

Review form: Reviewer 2

Recommendation

Major revision is needed (please make suggestions in comments)

Are each of the following suitable for general readers?

- a) **Title**
Yes
- b) **Summary**
No
- c) **Introduction**
No

Is the length of the paper justified?

No

Should the paper be seen by a specialist statistical reviewer?

No

Is it clear how to make all supporting data available?

Not Applicable

Is the supplementary material necessary; and if so is it adequate and clear?

Not Applicable

Do you have any ethical concerns with this paper?

No

Comments to the Author

This review is original as it covers a relevant and emerging topic about membrane transporters that has not been considered for a review article until now. The submitted article thus certainly fills a gap. But the review is very long, and it includes so many details that it is often difficult to read them extensively. For this reason, the review will be mostly useful to the small community of scientists directly involved in the study of transporter tails.

The authors should pay much more attention to the references they cite. I will not list here all the inappropriate citations I have found, as it would need too much work and time.

The first sentence of the abstract is problematic. It states that cell signaling is mediated by

transporters, which is obviously wrong. Some transporters indeed have signaling properties, but the proposed sentence has a different meaning.

I'm also intrigued by the first sentence of Introduction. The authors claim that the activity of transporters is essential for cell division and cell differentiation. What did the authors mean is unclear, as I don't know of any transporter shown to be essential for cell division. The authors also seem to ignore in this first sentence that small ions are frequent substrates of transporters. More generally, they should define more precisely what is the exact scope of the review, namely what is their definition of a transporter (are ion carriers excluded?), as this term is sometimes used by experts to qualify any transport protein including channels. In the next paragraphs, it appears that the authors focus on transport proteins undergoing conformational changes to translocate substrates across the membrane, excluding transport ATPases. This should be stated at the beginning of the review.

The review deals with transporters from different organisms, with a special emphasis on fungal transporters. But some general statements are only correct in the framework of a particular group of species. For instance, the sentence Recycling involves the maturation of post-endocytic early endosomes to recycling or sorting endosomes, and can also involve retrograde sorting to the trans-Golgi network (TGN), and from there re-routing back to the PM in conventional or transporter-specific Secretory Vesicles at p 11 is valid for mammalian cells, but certainly not for yeast (see eg. The recent work of B Glick published in 2018 in Dev Cell).

p. 5. This sentence is unclear: "This sequence proved to be redundant for transporter folding or function (Keener and Babst, 2013; Papadaki et al., 2018), .. What does "redundant" mean here ?

p. 7. In this sentence, is « other » necessary ? "Additionally to the presence of short sequence motifs, experimental evidence or ab initio structural predictions have suggested that the N- and C-termini of NSS contain partially conserved folds that seem to be extremely important for function, via their interaction with each other and other cytosolic loops of the main body of the NSS transporters "

p. 7, 8 : the authors don't use the common term "translocon", they rather use "translocase protein channel » and « translocate complex », which seem less commonly used and less accurate.

p. 8. For the sentence "... the ribosomes which translate transporters or other membrane proteins "need" to attach to the translocase complex in the ER .. " : ribosomes don't translate proteins, they translate mRNAs.

p. 8. ".. the COPII adaptor coat protein Sec24 or its paralogues .." : would « homologs » not be more appropriate ?

p. 9 : Regarding the sentence " Rather surprisingly, no mutation or condition has been shown to block PM transporters in the Golgi or in other post-ER compartment (secretory vesicles or endosomes). It seems that specific, de novo made, transporters, such as the CFTR channel involved in cystic fibrosis or the UapA purine transporter, exit the ER in COPII vesicles, but subsequent sorting steps involve an unconventional mechanism of trafficking that only requires proper, clathrin heavy chain-dependent, actin organization (Yoo et al., 2002; Rabouille, 2017; Bouris et al., 2019). ". The authors ignore important studies showing that some transporters undergo sorting regulation at Golgi level, with a role of cytosolic tails in this sorting. This aspect should also be covered by the review.

p. 9 : "Transporter ubiquitination is carried-out by HECT-type ubiquitin ligases of the Nedd4/Rsp5-type (Fang et al., 2014), recruited to transporter tails by adaptor proteins called α -

arrestins (Lin et al., 2008; Nikko and Pelham, 2009; MacGurn et al., 2012; Guiney et al., 2016; Mund and Pelham, 2018) .." The cited reference Fang et al., 2014 is inadequate, and regarding the adaptors, it would be fair to also cite the articles of S Leon and B Andre.

p. 17 : "What is however very interesting is that once these termini are "invented" during evolution, .. ". This kind of statement (« invented ») is unacceptable in a scientific article, even between brackets.

p. 19 : "Other reports have shown that lipids other than phospholipids (PIP), in particular sterols, seem to play important roles in the functioning of DAT .. " What does « PIP » mean here ?

p ; 20. "Recent crystal structures of DAT revealed the presence of two conserved cholesterol- like molecules and relative MD simulations suggested that outward-facing DAT in the absence of cholesterol DAT undergoes a conformational modification that mimics the transition to an inward-facing conformation." This sentence is not properly formulated.

References should be cited after these sentences:

P. 3 Unlike functional domains essential for transport activity, transporter termini .. in subcellular sorting, targeting, stability, cell-specific modification, or regulated turnover.

P.3 Same suggestion for the next sentence, in which an « of » should also be discarded.

P. 4 The additional two TMS are not involved directly in transport activity, and are believed to be essential for the oligomerization status of APCs in vivo,

P. 4 For example, the N-termini of yeast APCs possess relatively short motifs and specific Lys residues necessary for regulated turnover via ubiquitination, endocytic internalization, and sorting in endosomes and vacuoles.

P. 8 It is more than obvious that such lipid bilayer domains (Nie et al., 2018), processes also important for ER-exit,
And a dot should finish the sentence.

Decision letter (RSOB-19-0083.R0)

21-May-2019

Dear Professor Diallinas

We are pleased to inform you that your manuscript RSOB-19-0083 entitled "Tales of tails in transporters" has been accepted by the Editor for publication in Open Biology. The reviewer(s) have recommended publication, but also suggest some minor revisions to your manuscript. Therefore, we invite you to respond to the reviewer(s)' comments and revise your manuscript.

Please submit the revised version of your manuscript within 7 days. If you do not think you will be able to meet this date please let us know immediately and we can extend this deadline for you.

To revise your manuscript, log into <https://mc.manuscriptcentral.com/rsob> and enter your Author Centre, where you will find your manuscript title listed under "Manuscripts with

Decisions." Under "Actions," click on "Create a Revision." Your manuscript number has been appended to denote a revision.

- 1) A text file of the manuscript (doc, txt, rtf or tex), including the references, tables (including captions) and figure captions. Please remove any tracked changes from the text before submission. PDF files are not an accepted format for the "Main Document".
- 2) A separate electronic file of each figure (tiff, EPS or print-quality PDF preferred). The format should be produced directly from original creation package, or original software format. Please note that PowerPoint files are not accepted.
- 3) Electronic supplementary material: this should be contained in a separate file from the main text and meet our ESM criteria (see <http://royalsocietypublishing.org/instructions-authors#question5>). All supplementary materials accompanying an accepted article will be treated as in their final form. They will be published alongside the paper on the journal website and posted on the online figshare repository. Files on figshare will be made available approximately one week before the accompanying article so that the supplementary material can be attributed a unique DOI.

Online supplementary material will also carry the title and description provided during submission, so please ensure these are accurate and informative. Note that the Royal Society will not edit or typeset supplementary material and it will be hosted as provided. Please ensure that the supplementary material includes the paper details (authors, title, journal name, article DOI). Your article DOI will be 10.1098/rsob.2016[last 4 digits of e.g. 10.1098/rsob.20160049].

- 4) A media summary: a short non-technical summary (up to 100 words) of the key findings/importance of your manuscript. Please try to write in simple English, avoid jargon, explain the importance of the topic, outline the main implications and describe why this topic is newsworthy.

Images

Data-Sharing

It is a condition of publication that data supporting your paper are made available. Data should be made available either in the electronic supplementary material or through an appropriate repository. Details of how to access data should be included in your paper. Please see <http://royalsocietypublishing.org/site/authors/policy.xhtml#question6> for more details.

Data accessibility section

Sincerely,

The Open Biology Team
 mailto:openbiology@royalsociety.org

Reviewer(s)' Comments to Author:

Referee: 1

Comments to the Author(s)
 Diallinas and Mikros

Tales of tails in transporters

The manuscript reviews the state of the art with respect to the role of N and C-terminal regions in terms of transporter structure and function. These roles are emerging and the current paper, if edited somewhat, will be a useful addition to the current literature in this area. The figures are good quality and help to highlight key issues covered in the manuscript. My major issue with the manuscript is that it is far too long and relatively unfocused. There are lengthy descriptions of aspects of transporter biology that are not directly relevant to the key subject of the manuscript that distract from the topic being discussed. A substantial reduction in the length of the manuscript will make this easier to read and be more in keeping with the title of the paper.

In addition, the manuscript could do with an English edit.

Comments:

Page 2 Emphasis should be made at the start of the manuscript that the focus is secondary active transporters. A single mention of ABC transporters is made further down the page and this hardly seems relevant in this manuscript.

Page 2 Para 2 Line 22 Should really cite Forrest et al, 2011 here also.

Pages 3-4 Include a lengthy description of the different families of transporters. This could be cut down to just focus on the differences in the tail regions.

Pg 5 5th from last line. The text states that a specific motif might be critical for oligomerisation of UapA. What is the evidence for this?

Page 9-13 I think this could either be cut down or additional sections added

Page 13-14 Could dramatically cut down on the second paragraph in section 5. Only critical sentence is last one "Emerging evidence supports that the N- and ... "

MD is referred to throughout the manuscript as "showing" things. I would prefer the word "indicate"

Pg 16-17 Again another very long section. Should make more concise to focus on the points relevant to function of the tail region.

Pg 18-19 The section on oligomerisation has a very long introduction-could be summarised in a couple of sentences citing the Alguet and Cecchetti reviews.

Pg 19-20 The relevance of lipid interactions should be only the tail regions.

Pg 20 Last paragraph is very vague-should be really factual focusing on known material.

Referee: 2

Comments to the Author(s)

This review is original as it covers a relevant and emerging topic about membrane transporters that has not been considered for a review article until now. The submitted article thus certainly fills a gap. But the review is very long, and it includes so many details that it is often difficult to read them extensively. For this reason, the review will be mostly useful to the small community of scientists directly involved in the study of transporter tails.

The authors should pay much more attention to the references they cite. I will not list here all the inappropriate citations I have found, as it would need too much work and time.

The first sentence of the abstract is problematic. It states that cell signaling is mediated by transporters, which is obviously wrong. Some transporters indeed have signaling properties, but the proposed sentence has a different meaning.

I'm also intrigued by the first sentence of Introduction. The authors claim that the activity of transporters is essential for cell division and cell differentiation. What did the authors mean is unclear, as I don't know of any transporter shown to be essential for cell division. The authors also seem to ignore in this first sentence that small ions are frequent substrates of transporters. More generally, they should define more precisely what is the exact scope of the review, namely what is their definition of a transporter (are ion carriers excluded?), as this term is sometimes used by experts to qualify any transport protein including channels. In the next paragraphs, it appears that the authors focus on transport proteins undergoing conformational changes to translocate substrates across the membrane, excluding transport ATPases. This should be stated at the beginning of the review.

The review deals with transporters from different organisms, with a special emphasis on fungal transporters. But some general statements are only correct in the framework of a particular group of species. For instance, the sentence Recycling involves the maturation of post-endocytic early endosomes to recycling or sorting endosomes, and can also involve retrograde sorting to the trans-Golgi network (TGN), and from there re-routing back to the PM in conventional or

transporter-specific Secretory Vesicles at p 11 is valid for mammalian cells, but certainly not for yeast (see eg. The recent work of B Glick published in 2018 in Dev Cell).

p. 5. This sentence is unclear : "This sequence proved to be redundant for transporter folding or function (Keener and Babst, 2013; Papadaki et al., 2018), .. What does "redundant" mean here ?

p. 7. In this sentence, is « other » necessary ? "Additionally to the presence of short sequence motifs, experimental evidence or ab initio structural predictions have suggested that the N- and C-termini of NSS contain partially conserved folds that seem to be extremely important for function, via their interaction with each other and other cytosolic loops of the main body of the NSS transporters "

p. 7, 8 : the authors don't use the common term "translocon", they rather use "translocase protein channel » and « translocate complex », which seem less commonly used and less accurate.

p. 8. For the sentence "... the ribosomes which translate transporters or other membrane proteins "need" to attach to the translocase complex in the ER .. " : ribosomes don't translate proteins, they translate mRNAs.

p. 8. ".. the COPII adaptor coat protein Sec24 or its paralogues .." : would « homologs » not be more appropriate ?

p. 9 : Regarding the sentence " Rather surprisingly, no mutation or condition has been shown to block PM transporters in the Golgi or in other post-ER compartment (secretory vesicles or endosomes). It seems that specific, de novo made, transporters, such as the CFTR channel involved in cystic fibrosis or the UapA purine transporter, exit the ER in COPII vesicles, but subsequent sorting steps involve an unconventional mechanism of trafficking that only requires proper, clathrin heavy chain-dependent, actin organization (Yoo et al., 2002; Rabouille, 2017; Bouris et al., 2019). ". The authors ignore important studies showing that some transporters undergo sorting regulation at Golgi level, with a role of cytosolic tails in this sorting. This aspect should also be covered by the review.

p. 9 : "Transporter ubiquitination is carried-out by HECT-type ubiquitin ligases of the Nedd4/Rsp5-type (Fang et al., 2014), recruited to transporter tails by adaptor proteins called a-arrestins (Lin et al., 2008; Nikko and Pelham, 2009; MacGurn et al., 2012; Guiney et al., 2016; Mund and Pelham, 2018) .." The cited reference Fang et al., 2014 is inadequate, and regarding the adaptors, it would be fair to also cite the articles of S Leon and B Andre.

p. 17 : "What is however very interesting is that once these termini are "invented" during evolution, .. ". This kind of statement (« invented ») is unacceptable in a scientific article, even between brackets.

p. 19 : "Other reports have shown that lipids other than phospholipids (PIP), in particular sterols, seem to play important roles in the functioning of DAT .. " What does « PIP » mean here ?

p ; 20. "Recent crystal structures of DAT revealed the presence of two conserved cholesterol- like molecules and relative MD simulations suggested that outward-facing DAT in the absence of cholesterol DAT undergoes a conformational modification that mimics the transition to an inward-facing conformation." This sentence is not properly formulated.

References should be cited after these sentences:

P. 3 Unlike functional domains essential for transport activity, transporter termini .. in subcellular sorting, targeting, stability, cell-specific modification, or regulated turnover.

P.3 Same suggestion for the next sentence, in which an « of » should also be discarded.

P. 4 The additional two TMS are not involved directly in transport activity, and are believed to be essential for the oligomerization status of APCs in vivo,

P. 4 For example, the N-termini of yeast APCs possess relatively short motifs and specific Lys residues necessary for regulated turnover via ubiquitination, endocytic internalization, and sorting in endosomes and vacuoles.

P. 8 It is more than obvious that such lipid bilayer domains (Nie et al., 2018), processes also important for ER-exit,
And a dot should finish the sentence.

Author's Response to Decision Letter for (RSOB-19-0083.R0)

See Appendix A.

Decision letter (RSOB-19-0083.R1)

29-May-2019

Dear Professor Diallinas

We are pleased to inform you that your manuscript entitled "Tales of tails in transporters" has been accepted by the Editor for publication in Open Biology.

Sincerely,

The Open Biology Team
mailto: openbiology@royalsociety.org

Appendix A

NATIONAL AND KAPODISTRIAN
UNIVERSITY OF ATHENS
Department of Biology

To:
David M. Glover FRS
Editor in Chief
Royal Society's Open Biology

Dear David,

Please find our revised version of our review entitled “*Tales of tails in transporters*”, to be published in *Open Biology* of the Royal Society. The revision answers all points raised by the reviewers. Both wanted us to reduce the text and so we did. We cut out 2300 words which is a 23% reduction. Below you can see the point-to-point answers to other comments, mostly raised by reviewer 2. Thank you again for this invitation.

All the best

George

Referee: 1

Page 2 Emphasis should be made at the start of the manuscript that the focus is secondary active transporters. A single mention of ABC transporters is made further down the page and this hardly seems relevant in this manuscript.

Revised accordingly

Page 2 Para 2 Line 22 Should really cite Forrest et al, 2011 here also.

Reference added

Pages 3-4 Include a lengthy description of the different families of transporters. This could be cut down to just focus on the differences in the tail regions.

Reduced

Pg 5 5th from last line. The text states that a specific motif might be critical for oligomerisation of UapA. What is the evidence for this?

See Martzoukou et al, 2015. Evidence comes from BiFC assays, but also from pulldown assays and dominant negative effects.

Page 9-13 I think this could either be cut down or additional sections added

Reduced

Page 13-14 Could dramatically cut down on the second paragraph in section 5. Only critical sentence is last one “Emerging evidence supports that the N- and ... “

Revised accordingly

MD is referred to throughout the manuscript as “showing” things. I would prefer the word “indicate

Revised accordingly

Pg 16-17 Again another very long section. Should make more concise to focus on the points relevant to function of the tail region.

Reduced

Pg 18-19 The section on oligomerization has a very long introduction-could be summarised in a couple of sentences citing the Alguel and Cecchetti reviews.

Revised accordingly

Pg 19-20 The relevance of lipid interactions should be only the tail regions.

Revised accordingly

Pg 20 Last paragraph is very vague-should be really factual focusing on known material.

Revised accordingly

Referee: 2

The first sentence of the abstract is problematic. It states that cell signaling is mediated by transporters, which is obviously wrong. Some transporters indeed have signaling properties, but the proposed sentence has a different meaning.

Transporters can be essential in signaling, maybe not directly, but through the substrates they transport. In some cases, as stated later, they can also be direct mediators of signaling (e.g. transceptors), or sensors of the environment (the case of Aspergillus transporters during germination). We do not agree with the reviewer.

I'm also intrigued by the first sentence of Introduction. The authors claim that the activity of transporters is essential for cell division and cell differentiation. What did the authors mean is unclear, as I don't know of any transporter shown to be essential for cell division.

Again we do not agree with the reviewer. What we mean is that without proper nutrient, vitamin, etc., there is no cell division and in some cases in fungi no spore differentiation. Starvation or uptake of xenobiotics can elicit alterations in differentiation etc. In anyway, we removed cell division.

The authors also seem to ignore in this first sentence that small ions are frequent substrates of transporters. More generally, they should define more precisely what is the exact scope of the review, namely what is their definition of a transporter (are ion carriers excluded?), as this term is sometimes used by experts to qualify any transport protein including channels.

We are definitely aware that “ions are frequent substrates of transporters”. Anyway, we made clearer what kind of transporters this review discusses in the beginning of the revised version.

In the next paragraphs, it appears that the authors focus on transport proteins undergoing conformational changes to translocate substrates across the membrane, excluding transport ATPases. This should be stated at the beginning of the review.

Revised accordingly, see answer above.

The review deals with transporters from different organisms, with a special emphasis on fungal transporters. But some general statements are only correct in the framework of a particular group of species. For instance, the sentence Recycling involves the maturation of post-endocytic early endosomes to recycling or sorting endosomes, and can also involve retrograde sorting to the trans-Golgi network (TGN), and from there re-routing back to the PM in conventional or transporter-specific Secretory Vesicles at p 11 is valid for mammalian cells, but certainly not for yeast(see eg. The recent work of B Glick published in 2018 in Dev Cell

Text revised to make the reviewer’s point clear.

p. 5. This sentence is unclear : "This sequence proved to be redundant for transporter folding or function (Keener and Babst, 2013; Papadaki et al., 2018), ".. What does "redundant" mean here ?

It means that mutations in this motif can affect substrate specificity (enlarge it, allowing additional substrate to be recognized with low affinity), without affecting the kinetics of transport of the physiological substrates, which in turn means that it does not affect the overall folding of the transporter. This becomes clearer later, as indicated in the text.

p. 7. In this sentence, is « other » necessary? "Additionally to the presence of short sequence motifs, experimental evidence or ab initio structural predictions have suggested that the N- and C-termini of NSS contain partially conserved folds that seem to be extremely important for function, via their interaction with eachother and other cytosolic loops of the main body of the NSS transporters "

The word “other” deleted.

p. 7, 8 : the authors don’t use the common term “translocon”, they rather use “translocase protein channel » and « translocate complex, which seem less commonly used and less accurate.

“Translocon” used in the revised version

p. 8. For the sentence "... the ribosomes which translate transporters or other membrane proteins "need" to attach to the translocase complex in the ER ." :ribosomes don't translate proteins, they translate mRNAs.

Changed to "ribosomes which synthesizes"

p. 8. ".. the COPII adaptor coat protein Sec24 or its paralogues .." : would «homologs » not be more appropriate ?

Not really, as we wanted to emphasize that duplication of Sec24 has created paralogues that mght have different cargo specificities.

p. 9 : Regarding the sentence " Rather surprisingly, no mutation or condition has been shown to block PM transporters in the Golgi or in other post-ER compartment (secretory vesicles or endosomes). It seems that specific, de novo made, transporters, such as the CFTR channel involved in cystic fibrosis or the UapA purine transporter, exit the ER in COPII vesicles, but subsequent sorting steps involve an unconventional mechanism of trafficking that only requires proper, clathrin heavy chain-dependent, actin organization (Yoo et al., 2002; Rabouille, 2017; Bouris et al., 2019). ". The authors ignore important studies showing that some transporters undergo sorting regulation at Golgi level, with a role of cytosolic tails in this sorting. This aspect should also be covered by the review.

We definitely do not ignore this issue but we have a different opinion. As this has been a big issue for our lab, we have extensively reviewed the literature and found that in no case formal proof exists show Jung that de novo made transporter pass through the Golgi. Transporter sorting to the Golgi seems to concern transporter recycling back from the PM. This, we believe, has caused a general misinterpretation concerning the use of conventional secretory path for transport secretion. Indeed, we have performed multiple experiments and showed that several Aspergillus transporters use a non-conventional secretory route. This can be seen in a preprint of our work at BioRxiv (Bouris et al. 2019). Actually we are revising our work to be resubmitted to EMBO J. We understand that this is a rather provoking finding that it is hard others to buy it. We are doing our best to convince our colleagues that transporters follow an unconventional path and we insist of presenting this concept in this review. The whole subject of transporter sorting is however little related to the roles of tails so no more information on the different paths is needed here.

p. 9 : "Transporter ubiquitination is carried-out by HECT-type ubiquitin ligases of the Nedd4/Rsp5-type (Fang et al., 2014), recruited to transporter tails by adaptor proteins called α -arrestins (Lin et al., 2008; Nikko and Pelham, 2009; MacGurn et al., 2012; Guiney et al., 2016; Mund and Pelham, 2018)" The cited reference Fang et al., 2014 is inadequate, and regarding the adaptors, it would be fair to also cite the articles of S Leon and B Andre.

The reviewer is absolutely right; we missed form the text articles of the two most important groups of this subject. Added in the revised version.

p. 17 : "What is however very interesting is that once these termini are "invented" during evolution, ." . This kind of statement (« invented ») is unacceptable in ascientific article, even between brackets.

Replaced with "created"

p. 19 : "Other reports have shown that lipids other than phospholipids (PIP), in particular sterols, seem to play important roles in the functioning of DAT .. " What does « PIP » mean here ?

Removed

p ; 20. "Recent crystal structures of DAT revealed the presence of two conserved cholesterol-like molecules and relative MD simulations suggested that outward-facing DAT in the absence of cholesterol DAT undergoes a conformational modification that mimics the transition to an inward-facing conformation." This sentence is not properly formulated.

This sentence was rephrased to become clearer.

References should be cited after these sentences:

P. 3 Unlike functional domains essential for transport activity, transporter termini.. in subcellular sorting, targeting, stability, cell-specific modification, or regulated turnover.

References added

P.3 Same suggestion for the next sentence, in which an « of » should also be discarded.

Corrected

P. 4 The additional two TMS are not involved directly in transport activity, and are believed to be essential for the oligomerization status of APCs in vivo,

Reference added

P. 4 For example, the N-termini of yeast APCs possess relatively short motifs and specific Lys residues necessary for regulated turnover via ubiquitination, endocytic internalization, and sorting in endosomes and vacuoles.

References added

P. 8 It is more than obvious that such lipid bilayer domains (Nie et al., 2018), processes also important for ER-exit, And a dot should finish the sentence.

Corrected